# FAK Executes Anti-Senescence via Regulating EZH2 Signaling in Non-Small Cell Lung Cancer Cells

**DOI:** 10.3390/biomedicines10081937

**Published:** 2022-08-10

**Authors:** Hsiang-Hao Chuang, Ming-Shyan Huang, Yen-Yi Zhen, Cheng-Hao Chuang, Ying-Ray Lee, Michael Hsiao, Chih-Jen Yang

**Affiliations:** 1Division of Pulmonary Critical Care Medicine, Department of Internal Medicine, Kaohsiung Medical University Hospital, Kaohsiung Medical University, Kaohsiung 80708, Taiwan; 2Department of Internal Medicine, E-Da Cancer Hospital, School of Medicine, I-Shou University, Kaohsiung 82445, Taiwan; 3Division of Nephrology, Department of Internal Medicine, Kaohsiung Medical University Hospital, Kaohsiung Medical University, Kaohsiung 80708, Taiwan; 4Department of Microbiology and Immunology, College of Medicine, Kaohsiung Medical University, Kaohsiung 80708, Taiwan; 5Genomics Research Center, Academia Sinica, Taipei 11529, Taiwan; 6Faculty of Post-Baccalaureate Medicine, College of Medicine, Kaohsiung Medical University, Kaohsiung 80708, Taiwan; 7Cancer Center, Kaohsiung Medical University Hospital, Kaohsiung Medical University, Kaohsiung 80708, Taiwan

**Keywords:** FAK, EZH2, senescence, lamin A/C

## Abstract

Focal adhesion kinase (FAK) is a non-receptor tyrosine kinase overexpressed in various cancer types that plays a critical role in tumor progression. Accumulating evidence suggests that targeting FAK, either alone or in combination with other agents, may serve as an effective therapeutic strategy for numerous cancers. In addition to retarding proliferation, metastasis, and angiogenesis, FAK inhibition triggers cellular senescence in lung cancer cells. However, the detailed mechanism remains enigmatic. In the present study, we found that FAK inhibition not only elicits DNA-damage signaling but also downregulates enhancer of zeste homolog 2 (EZH2) expression. The manipulation of FAK expression influences EZH2 expression and corresponding signaling in vitro. Immunohistochemistry shows that active FAK signaling corresponds with the activation of the EZH2-mediated signaling cascade in lung-cancer-cells-derived tumor tissues. We also found that ectopic EZH2 expression attenuates FAK-inhibition-induced cellular senescence in lung cancer cells. Our results identify EZH2 as a critical downstream effector of the FAK-mediated anti-senescence pathway. Targeting FAK-EZH2 axis-induced cellular senescence may represent a promising therapeutic strategy for restraining tumor growth.

## 1. Introduction

Lung cancer is one of the deadliest malignancies worldwide, contributing to approximately 1.8 million deaths in 2020 [1]. Despite advancements in therapeutic strategies, lung cancer prognosis remains poor, with tumor metastasis and recurrence representing major contributors to lung-cancer-associated mortality [2,3,4].

Focal adhesion kinase (FAK) is a non-receptor tyrosine kinase composed of multiple domains that regulate diverse cellular processes, including growth factor signaling, cell cycle progression, cell survival, cell motility, angiogenesis, and the establishment of an immunosuppressive tumor microenvironment (TME), through kinase-dependent and -independent scaffolding functions in both the cytoplasm and nucleus [5]. Increasing evidence supports a role for FAK in malignant processes, and FAK is considered a promising pharmaceutical target for anticancer therapies [6,7].

In addition to roles in cell survival and proliferation, FAK-mediated signaling also inhibits senescence programs. Increasing evidence shows that targeting FAK promotes cellular senescence in cancer cells [8,9,10]. FAK inhibition results in nuclear deformity and induces the DNA-damage response that accompanies cellular senescence. However, the detailed mechanism through which FAK inhibition results in the activation of senescence programs remains enigmatic.

Enhancer of zeste homolog 2 (EZH2), together with embryonic ectoderm development (EED) and SUZ12, comprises polycomb repressive complex 2 (PRC2), which methylates Lys9 and Lys27 on histone H3, leading to the transcriptional repression of downstream genes. EZH2 has been identified as a critical downstream component of the retinoblastoma tumor suppressor protein (pRB)–E2F pathway, which is essential for cancer cell proliferation and growth [11]. EZH2 is an epigenetic regulator of cell proliferation that is overexpressed in most cancers, and high levels of EZH2 expression correlate with poor prognosis [12,13,14,15,16]. In addition, EZH2 also plays a critical role in anti-senescence programs in both non-tumor cells and tumor cells. EZH2 depletion activates p16 and p21, which triggers cellular senescence in mouse embryonic fibroblasts and melanoma cells [17,18]. Therefore, EZH2 is regarded as a potential new target for cancer treatment [19].

In the present study, we found that FAK inhibition reduced EZH2 expression. The manipulation of FAK expression using knockdown or overexpression approaches influenced EZH2-expression levels. Furthermore, the overexpression of EZH2 attenuated FAK-inhibition-induced cellular senescence in non-small cell lung cancer cells. It implies that EZH2 might be a critical downstream effector of FAK-mediated anti-senescence signaling. It also reveals a promising target, the FAK–EZH2 axis, for cancer treatment.

## 2. Materials and Methods

### 2.1. Materials

Detailed information regarding the materials used in this study can be found in Appendix A.

### 2.2. Cell Culture

A549 and H1299 cells were purchased from the American Type Culture Collection (ATCC). Both cell lines were cultured in RPMI 1640 medium, supplemented with 10% fetal bovine serum (Gibco, Grand Island, NY, USA), penicillin, and streptomycin (100 U/mL). All cells were maintained at 37 °C in a humidified incubator with a 5% CO_2_ atmosphere.

### 2.3. Western Blot Analysis

Immunoblotting was performed as described in our previous study [20]. The cells were harvested and lysed in 1× radioimmunoprecipitation assay (RIPA) buffer containing protease and phosphatase inhibitors. Protein concentrations were determined by a Bradford protein assay using the Bio-Rad DC protein kit (Bio-Rad Laboratories, Hercules, CA, U.S.). Equal amounts (30 μg) of total cell lysate were separated by 7–15% sodium dodecyl sulfate-polyacrylamide gel electrophoresis, followed by transfer to a polyvinylidene difluoride membrane. Protein expression was examined using primary antibodies, followed by horseradish peroxidase-conjugated secondary antibodies. Protein bands were detected with enhanced chemiluminescence (GE Healthcare, Chicago, IL, USA).

### 2.4. Tissue Microarray Construction

The biopsies were formalin-fixed, and the specimens were embedded in paraffin blocks. The present study was approved by the Institutional Review Board of Kaohsiung Medical University Hospital, and all subjects consented and signed informed consent forms for immunohistochemical examinations. In the present study, immunohistochemistry (IHC) imaging also included paired normal lung parenchymal tissues and tumor tissues. For animal study, all animal experiments were performed in accordance with a protocol approved by the Institutional Animal Care and Utilization Committee at Academia Sinica. The tumor tissues were collected from severe combined immune deficiency mutation and IL2 receptor gamma chain deficiency (NOD/SCID gamma) male mice with the subcutaneous implantation of indicated cancer cells (1 × 10^4^ cells in PBS with matrigel) into the left flank.

### 2.5. Immunohistochemistry (IHC)

The procedure used for immunohistochemical staining was described in our previous study [21]. Specimens were dewaxed in xylene twice for 10 min each and rehydrated for 5 min each in a graded series of ethanol-aqueous solutions (100%, 95%, and 75%). The specimens were soaked in 3% H_2_O_2_ in methanol for 15 min to inactivate endogenous peroxidase. To perform antigen retrieval, the specimens were treated in gently boiling citrate buffer (pH 6.0) for 15 min. The specimens were then blocked using 10% normal goat serum in Tris-buffered saline. To visualize the desired antigens, antibodies against EZH2, H3K27me3, and phosphorylated FAK were applied overnight at 4 °C in a humidified container. The distributions of EZH2, H3K27me3, and p-FAK in cancer tissues were visualized using commercially available peroxidase IHC kits and developed with diaminobenzidine (DAB; ThermoFisher, Waltham, MA, USA). Histochemical and immunohistochemical staining was scanned using a microscope and digitalized with a Pannoramic MIDI digital scanner (3DHISTECH Ltd., Budapest, Hungary). Microscopic photos were acquired using CaseViewer 2.3 (3DHISTECH Ltd., Budapest, Hungary).

### 2.6. Senescence-Associated β-Galactosidase Staining (SA-β-Gal Staining)

Detailed experimental procedures for SA-β-gal staining are described in our previous study [8]. The cells were fixed in aqueous 4% paraformaldehyde in 1 × PBS for 20 min. After fixation, the enzymatic activity of β-galactosidase activity was estimated by applying a senescent assay mixture comprised of substrate and buffer [1 mg/mL 5-bromo-4-chloro-3-indolyl β D-galactopyranoside (X-gal), 5 mM K_3_Fe (CN)_6_, 5 mM K_4_Fe (CN)_6_, 2 mM MgCl_2_, 150 mM NaCl, 40 mM citric acid, and 40 mM Na_2_HPO_4_ at pH 6.0] in the dark at 37 °C for 16 h. SA-β-gal activity was measured as the proportion of hydrolyzed X-gal, which produces a blue precipitate.

### 2.7. Statistical Analysis

All statistical analyses were performed using the two-tailed Student’s t-test. The data in this study are presented as the mean ± standard deviation of at least three independent experiments. * *P* < 0.05, ** *P* < 0.01, and *** *P* <0.001 indicate significant differences among the experimental groups.

## 3. Results

### 3.1. FAK Signaling Regulates EZH2 Expression and Function

Accumulating evidence has indicated that FAK may serve as a promising pharmaceutical target for anticancer therapies. Our previous study revealed that FAK inhibition triggers lamin A/C downregulation-mediated nuclear deformity and DNA-damage response accompanied by cellular senescence [8]. However, the detailed mechanisms through which FAK inhibition initiates these changes remain incompletely understood. FAK inhibitor treatment leads to the development of extremely large and deformed nuclei in lung cancer cells, suggesting that FAK inhibition triggers epigenetic alterations that promote cellular senescence. Baell et al. found that the inhibition of histone acetyltransferases KAT6A/B induces senescence and arrests tumor growth [22]. Therefore, we explored whether FAK inhibition induces cellular senescence through the KAT6A/B inhibition-mediated pathway. We treated lung cancer cells with WM-1119, a specific KAT6A/B inhibitor, and examined whether KAT6A/B inhibition induces lamin A/C downregulation. However, WM-1119 treatment did not downregulate lamin A/C (Appendix A), suggesting that FAK inhibition induces cellular senescence through distinct pathways.

EZH2 is a critical epigenetic regulator involved in cell proliferation that is known to prevent cellular senescence in cancer cells via the inhibition of cyclin-dependent kinase inhibitors. We explored whether FAK inhibition induces cellular senescence through an EZH2 inhibition-mediated pathway by treating lung cancer cells with a FAK inhibitor, PF-573228, and examining whether FAK inhibition affects EZH2 expression or function. The phosphorylation of FAK at Tyr-397 (p-FAK) represents the enzymatic activation of FAK [23]. To assess FAK kinase activity and verify FAK inhibition by PF-573228, p-FAK was detected using a specific antibody, which showed that FAK activity decreased over time following treatment with 10 μM PF-573228 in A549 and H1299 cells (Figure 1). We also found that the overall level of FAK increased slightly with PF-573228 treatment, which might represent an attempt to compensate for FAK inactivation. In addition to FAK inactivation, PF-573228 treatment reduced EZH2 expression (Figure 1), suggesting that FAK signaling may be involved in the regulation of EZH2 expression. PF-573228 treatment also upregulated p53, p21, and p27 expression in A549 and H1299 (p53 null) cells (Figure 1). This implies FAK inhibition induces DNA-damage signaling pathways.

We next explored whether the manipulation of FAK expression influences EZH2 expression. Compared with a GFP-expressing group, cells overexpressing GFP–FAK presented with slightly increased EZH2 levels in A549 and H1299 cells (Figure 2A). We also examined whether FAK depletion affects EZH2 expression. FAK knockdown was realized through the ectopic expression of small hairpin RNA (shRNA) vectors in H1299 cells. FAK downregulation reduced EZH2 expression, accompanied by a decrease in the trimethylation of histone H3 on Lys9 and Lys27 (Figure 2B), indicating that FAK inhibition reduces EZH2-mediated signaling.

### 3.2. Expression of EZH2, H3K27me3, and p-FAK in Lung-Cancer-Cells-Derived Tumor Tissues

The results of our experiments indicated that FAK activity might regulate EZH2-mediated signaling in lung cancer cells. To investigate the pre-clinical relevance of active FAK and EZH2-mediated signaling in vivo, we performed IHC analysis to detect the protein levels of p-FAK, EZH2, and its corresponding substrate, H3K27me3, in lung-cancer-cells-derived tumor tissue arrays. We detected high levels of p-FAK expression in regions with high levels of EZH2 and H3K27me3 expression in lung-cancer-cells-derived tumor tissues but not in the normal tissue (Figure 3). Although EZH2 and H3K27me3 were largely localized to the nucleus, whereas p-FAK was primarily localized to the cytoplasm, the staining regions coincided, indicating that FAK signaling correlates with EZH2 signaling.

### 3.3. EZH2 Depletion Promotes Cellular Senescence

Our experimental results suggested that EZH2 might serve as a downstream effector of FAK signaling. We examined whether EZH2 acts as a critical regulator of a master anti-senescence signaling pathway in non-small cell lung cancer cells. The SA β-gal staining assay was used to measure the effects of EZH2 depletion on cellular senescence in A549 and H1299 cells. SA-β-gal-positive cells represented <4% of the total A549 and H1299 cell population expressing shLuc compared with >10% and 12% of A549 and H1299 cells, respectively, expressing shEZH2, indicating that EZH2 depletion induced cellular senescence in non–small cell lung cancer cells (Figure 4A). EZH2 knockdown also downregulated lamin A/C expression (Figure 4B). It is consistent with our previous study showing that FAK inhibition mediated lamin A/C downregulation [8].

### 3.4. EZH2 Overexpression Attenuates FAK-Inhibition-Induced Cellular Senescence in Non-Small Cell Lung Cancer Cells

To further verify the critical role of EZH2 in FAK-mediated anti-senescence signaling, we explored whether EZH2 overexpression bypasses FAK-inhibition-induced cellular senescence in non-small cell lung cancer cells. In the vector control group (myc-His), 4 days of treatment with the FAK inhibitor induced cellular senescence, resulting in approximately 30% and 50% of A549 and H1299 cells showing SA-β-gal-positive staining (Figure 5). EZH2 overexpression significantly suppressed PF-573228-treatment-induced cellular senescence in A549 and H1299 cells (Figure 5), indicating that EZH2 acts as a critical downstream effector of FAK-mediated anti-senescence programming in non-small cell lung cancer cells.

## 4. Discussion

FAK regulates diverse biological processes involved in tumorigenesis, including proliferation, metastasis, angiogenesis, and the establishment of immunosuppressive tumor microenvironments, and targeting FAK is a potentially promising cancer treatment strategy. In addition to cell survival, proliferation, and metastasis, FAK-mediated mechanotransduction transmits extracellular mechanical cues to the cellular transcriptional machinery, regulating various biological processes [24]. FAK inhibition induces lamin A/C downregulation, followed by nuclear deformity and epigenetic alterations. Epigenetic mechanisms have now emerged as key contributors to the alterations of genome structure and function that accompany cellular senescence. The three pillars of epigenetic regulation are DNA methylation, histone modifications, and noncoding RNA species. Mounting evidence suggests that histone modifications influence cellular senescence programs [25,26]. Baell et al. found that the inhibition of histone acetyltransferases KAT6A/B induces senescence and suppresses tumor growth [22]. Treatment with WM-1119, a KAT6A/B inhibitor, induced cell cycle exit and cellular senescence without causing DNA damage, lamin A/C downregulation, or cell death. KAT6A/B inhibition suppresses the entrance of cells into the S phase, whereas FAK inhibition arrests cells in the G2/M phase. Therefore, KAT6A/B inhibition and FAK inhibition appear to induce cellular senescence through distinct pathways.

In addition to histone acetylation, the regulation of histone methylation also plays a critical role in epigenetic-alteration-mediated cellular senescence [25,26]. EZH2, a histone methyltransferase, is commonly overexpressed in most cancers and correlates with poor prognosis. Increasing evidence shows that EZH2 depletion induces cellular senescence through the activation of p53 and cyclin-dependent kinase inhibitors such as p16, p21, and p27 [17,18,27,28,29]. In our study, we found that FAK inhibition resulted in the downregulation of EZH2 and the upregulation of p53, p21, and p27, accompanied by cellular senescence in lung cancer cell lines. Due to the double-deleted and highly methylated p16 gene locus in A549 and H1299 cells, respectively, we did not observe p16 upregulation following FAK inhibition. In other cell types, FAK inhibition may induce p16 expression, resulting in potential cell cycle arrest and even senescence [30]. Furthermore, FAK overexpression results in only a slight increase in EZH2 expression in A549 and H1299 cells. It is similar to the contribution of FAK signaling to lamin A/C expression. FAK inhibition reduces lamin A/C expression, but FAK overexpression has no effect on lamin A/C expression (Appendix A). Interestingly, Gnani et al. found that FAK depletion induces DNA-damage signaling and reduces EZH2 expression and activity in hepatocellular carcinoma cells, which is consistent with our present study findings [31].

In our previous study, FAK inhibition triggers lamin A/C downregulation and nuclear deformity [8]. Li et al. showed that EZH2 knockdown induces caspase-mediated lamin A cleavage-driven apoptosis in SKOV3 cells [32]. We found that PF-573228 treatment induces the cleavage of lamin A/C (Appendix A) and certain cell death [8]. It implies that a FAK–EZH2 axis regulates cell survival and anti-senescence functions (Figure 5). The ectopic expression of EZH2 alleviates FAK-inhibitor-induced cellular senescence in non-small cell lung cancer cells (Figure 4), further supporting the hypothesis that the FAK–EZH2 axis regulates senescence programming. Moreover, our IHC analysis revealed that the EZH2-mediated signaling cascade is spatially correlated with active FAK signaling in lung cancer tissues and cell lines (Figure 3). Zhou et al. showed that the overexpression of EZH2 and FAK was associated with poor prognosis in high-grade endometrial carcinoma [33]. Furthermore, Gnani et al. reported the concomitant upregulation of FAK, EZH2, and H3K27me3 in hepatocellular carcinoma tissue arrays, which was correlated with disease severity, as determined by tumor staging, differentiation, and metastasis [31]. In conclusion, our data reveal that the FAK–EZH2 axis acts as an anti-senescence pathway, indicating that targeting this pathway may serve as a promising therapeutic strategy for cancer treatment (Figure 6).

Targeting FAK-induced cellular senescence as a promising cancer therapy might be applied to treat cancer diseases. Subsequently, the senescent cells activate the senescence-associated secretory phenotype (SASP) to release a collection of proinflammatory factors [34]. Therefore, several approaches have been built to pharmacologically develop compounds that target the senescent cells and SASP. Pharmacological strategies selectively eliminating senescent cells (senolytics) or confining its development (senomorphics) are considered [35]. A new field, known as senotherapeutics, has developed for healthy aging and age-related diseases, including cancers [35,36,37,38]. Chemical compounds targeting senescent cells are regarded as an attractive strategy to achieve antitumor treatment [39]. However, there is compelling evidence that shows that senescent cells can elicit local and systemic inflammation and promote tumorigenesis [35,40]. Mechanistic investigation reveals that many cancer interventions such as genotoxic chemotherapies induce DNA damage and promote senescence in cancer and non-malignant cells. The SASP from surrounding senescent cells has been shown to facilitate tumorigenesis [40,41,42]. FAK-inhibition-induced cellular senescence was observed in cancer cells but not in normal cells [8]. Alternatively, FAK inhibition promotes immune surveillance by reducing the fibrotic and immunosuppressive tumor microenvironment and renders tumors sensitive to checkpoint immunotherapy [43]. In conclusion, targeting the FAK–EZH2 axis represent a promising therapeutic strategy for various cancers.

## 5. Conclusions

Solid tumors are complex structures, which consists of a complicated heterogeneous mixture of tumor, immune, and stromal cells. FAK signaling facilitates cell proliferation and drug resistance in tumor cells and the establishment of immunosuppressive TME. Furthermore, overexpression of FAK is a malignant feature of various cancers, correlated with poor prognoses in numerous cancer patients. Therefore, FAK is an attractive therapeutic target for cancer treatment. In the present study, we identified that FAK promotes cell proliferation and anti-senescence partially through strengthening EZH2-mediated signaling. We propose that targeting FAK-induced cellular senescence or interruption of multiple oncogenic processes might represent a promising therapeutic strategy or an adjuvant one to promote the efficacy of combination therapies for cancer treatment.

## Figures and Tables

**Figure 1 biomedicines-10-01937-f001:**
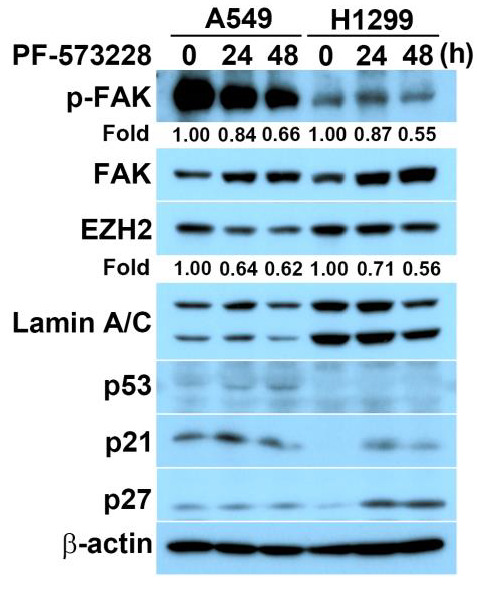
FAK inhibition reduces EZH2 expression and elicits DNA-damage signaling. A549 or H1299 cells were exposed to 10 μM PF-573228 for the indicated time, followed by lysate harvesting. It was subjected to immunoblotting for the indicated proteins. As a loading control, an anti-β-actin antibody was used. The numbers below the blot images indicate the relative intensity of pY397 FAK or EZH2 signal relative to the β-actin signal.

**Figure 2 biomedicines-10-01937-f002:**
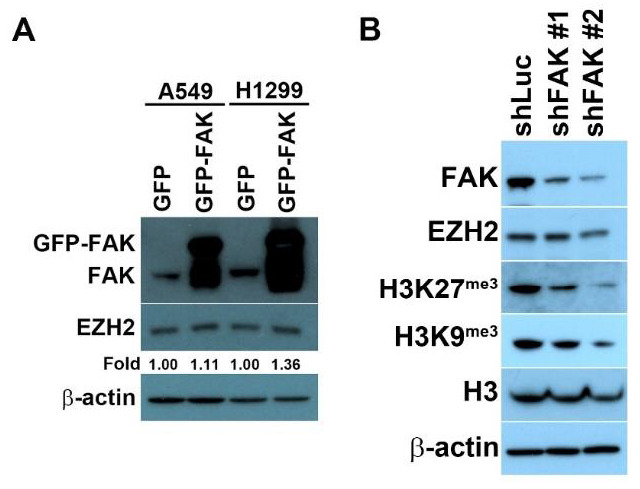
The manipulation of FAK expression regulates the functional roles of EZH2. (**A**) A549 or H1299 cells overexpressing GFP or GFP–FAK were lysed and subjected to Western blot analysis. The protein levels were detected using the indicated antibodies. A slight increase in EZH2 levels was detected in cells overexpressing GFP–FAK. (**B**) H1299 cells transfected with shFAK for FAK depletion were lysed and subjected to Western blot analysis. Protein levels were detected using the indicated antibodies. As a loading control, an anti-β-actin antibody was used. Western blot analysis revealed that FAK depletion decreased EZH2 expression and reduced downstream signaling, based on the assessment of the tri-methylation at Lys9 and Lys27 on histone H3.

**Figure 3 biomedicines-10-01937-f003:**
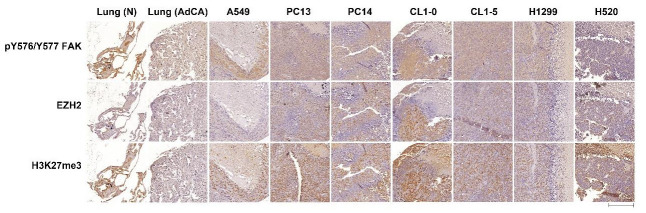
Immunohistochemistry of tissue microarrays reveals the functional relevance between FAK and EZH2 that exists in human non-small cell lung-cancer-cells-developed tumor tissues. Tissue microarray sections were subjected to immunostaining using anti-EZH2, -H3K27me3, and -phospho-FAK antibodies. Comparison of EZH2, H3K27me3, and phospho-FAK expression in normal tissues and tumor tissues from the tumor tissue microarray. (Bar scale: 200 μm).

**Figure 4 biomedicines-10-01937-f004:**
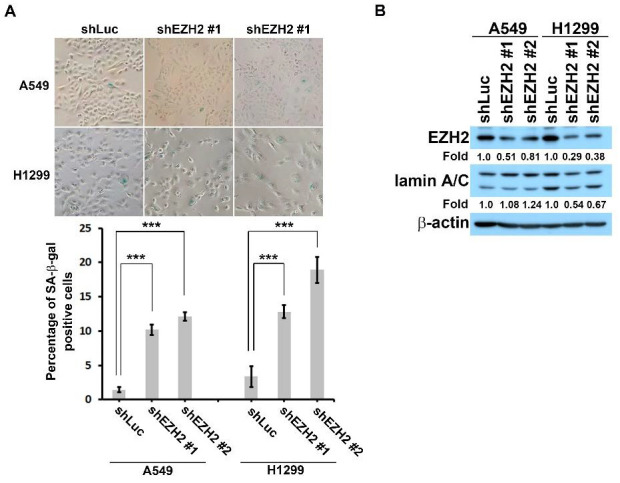
EZH2 knockdown induces cellular senescence in non-small cell lung cancer cells. (**A**) A549 or H1299 cells were transfected with shRNA to deplete EZH2 and incubated for 4 days. SA-β-Gal staining was employed to measure the percentage of senescent cells (β-gal-positive cells). The bar chart shows that <3% of cells in the shLuc population were SA-β-gal-positive, whereas more than 10% of the shFAK cells population were SA-β-gal positive, based on three independent experiments (*n* > 250). *** *p* < 0.001 according to a Student’s t-test. (**B**) A549 or H1299 cells were treated with shRNA to deplete EZH2, lysed, and subjected to Western blot analysis. Protein levels were detected using the indicated antibodies. As a loading control, an anti-β-actin antibody was used. The numbers below the blot images indicate the relative intensity of EZH2 or lamin A/C signal relative to the β-actin signal.

**Figure 5 biomedicines-10-01937-f005:**
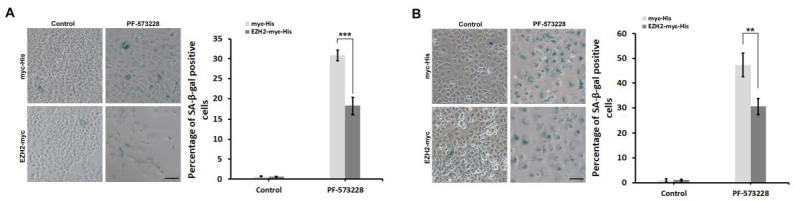
EZH2 overexpression attenuates FAK-inhibitor-induced cellular senescence in non-small cell lung cancer cells. (**A**) A549 or (**B**) H1299 cells overexpressing myc-His or EZH2-myc-His were exposed to 0 or 10 μM PF-573228 for 4 days. SA-β-Gal staining was employed to measure the percentage of senescent cells (β-gal-positive cells). The ratio of SA-β-gal-positive cells to total cells was calculated and plotted in a bar chart. The data represent the mean ± SD from three independent experiments (*n* > 250). ** *p* < 0.01, *** *p* < 0.001 according to a Student’s t-test. (Bar scale: 200 μm).

**Figure 6 biomedicines-10-01937-f006:**
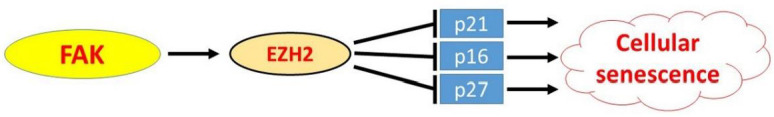
FAK executes anti-senescence via regulating EZH2 signaling. A proposed mechanism shows that FAK suppresses cellular senescence through strengthening the EZH2-mediated repression of cyclin-dependent kinase inhibitors in non-small cell lung cancer cells.

## Data Availability

Not applicable.

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
