# Peer review of "FAK Executes Anti-Senescence via Regulating EZH2 Signaling in Non-Small Cell Lung Cancer Cells"

_biomedicines, 2022, doi:10.3390/biomedicines10081937_

Round 1

Reviewer 1 Report

In this manuscript, Chuang et. al. focused on FAK and EZH2, two anti-cancer drug targets, and performed multiple assays including RNAi, IHC, etc. in cancer cells. The results revealed a FAK-EZH2 axis serving as a signaling pathway for anti-senescence. The manuscript has clear logic and sufficient data. However, several concerns should be addressed before accepted for publication.

Major points

1.     Page 3 Line129, “data not shown”. Any conclusion or claim should be supported by data. Even negative data are worth to shown. Please include this data, maybe in a supplementary figure.

2.     In Figure 1, the western blot of p27 shows two adjacent bands. Did the authors use other experiment to confirm which band is p27?

3.     In Figure 3, please add a clear scale bar to the figure.

4.     In Figure 4B, the lamin A/C bands in A549 cell line shows no significant difference by eye. Please show the relative intensity to β-actin (as in Figure 1). 

Minor points

1.     Page 1 Line 26-27, please correct this sentence “the detailed mechanism still detail mechanism remains enigmatic…”

2.     Overall, the writing should be improved.

Author Response

Comments:

Reviewer 1:

Comments and Suggestions for Authors:

In this manuscript, Chuang et. al. focused on FAK and EZH2, two anti-cancer drug targets, and performed multiple assays including RNAi, IHC, etc. in cancer cells. The results revealed a FAK-EZH2 axis serving as a signaling pathway for anti-senescence. The manuscript has clear logic and sufficient data. However, several concerns should be addressed before accepted for publication.

Major points

  1. Page 3 Line129, “data not shown”. Any conclusion or claim should be supported by data. Even negative data are worth to shown. Please include this data, maybe in a supplementary figure.

Response: We are grateful for your kind advice. We add the data which KAT6A/B inhibition does not influence lamin A/C expression in the supplementary figure 1 in the revised manuscript.

  1. In Figure 1, the western blot of p27 shows two adjacent bands. Did the authors use other experiment to confirm which band is p27?

Response: We appreciate the professional comments of the reviewer. Due to lack of an UV crosslinker device, we treated cells with aphidicolin, a specific inhibitor of DNA polymerase Alpha and Delta, to activate replication stress-induced DNA damage signaling. Although aphidicolin treatment did not induce p27 expression but activates p53 and p21 signaling. Therefore, we speculate the other band under 25 kDa is a non-specific band.

  1. In Figure 3, please add a clear scale bar to the figure.

Response: We are grateful for your kind advice. We add a clear scale bar in Figure 3 in the revised manuscript.

  1. In Figure 4B, the lamin A/C bands in A549 cell line shows no significant difference by eye. Please show the relative intensity to β-actin (as in Figure 1).

Response: We appreciate the professional comments of the reviewer. We showed the relative intensities of EZH2 and lamin A/C to β-actin in the revised manuscript. EZH2 knockdown does not trigger downregulation of lamin A/C in A549 cells (left part of Figure 4B). It might be due to the time point or different cellular contexts in A549 and H1299 cells. However, EZH2 knockdown promotes cellular senescence in A549 and H1299 cells through lamin A/C downregulation-mediated pathway or distinct pathways.

Minor points

  1. Page 1 Line 26-27, please correct this sentence “the detailed mechanism still detail mechanism remains enigmatic…”

Response: We apologize for the carelessness and appreciate the kind comment. We correct the fault in the revised manuscript.

  1. Overall, the writing should be improved.

Response: We appreciate the helpful comment of the reviewer. The manuscript is proofread by the language editing service. The present manuscript is revised.

Reviewer 2 Report

In the manuscript entitled "FAK executes anti-senescence via regulating EZH2 signaling in non-small cell lung cancer cells"  have identified FAK-EZH2 axis acts as a signaling pathway for anti-senescence which could be potentially used as a therapeutic strategy for cancer treatment. The authors have very well designed the experiments and the results are presented well. Please consider the following suggestion to improve the rigor of the manuscript.

1. Line 26 needs to be rewritten for clarity.

2. Line 30 "In the meanwhile,...." needs to be modified for clarity.

3. In general the abstract can be modified to provide simplicity and effective understanding to the reader.

4. Please consider the correction of grammatical errors and language clarity throughout the manuscript. 

5. line 156, change 'companied' to 'accompanied'

6. Please add some discussion about future perspectives to the findings obtained from the studies in the discussion/conclusion section

Author Response

Reviewer 2:

Comments and Suggestions for Authors:

In the manuscript entitled "FAK executes anti-senescence via regulating EZH2 signaling in non-small cell lung cancer cells" have identified FAK-EZH2 axis acts as a signaling pathway for anti-senescence which could be potentially used as a therapeutic strategy for cancer treatment. The authors have very well designed the experiments and the results are presented well. Please consider the following suggestion to improve the rigor of the manuscript.

  1. Line 26 needs to be rewritten for clarity.

Response: We apologize for the carelessness. We correct the fault in the revised manuscript.

  1. Line 30 "In the meanwhile,...." needs to be modified for clarity.

Response: We appreciate the kind comment of the reviewer. We correct the ambiguous description in the revised manuscript.

  1. In general the abstract can be modified to provide simplicity and effective understanding to the reader.

Response: We appreciate the helpful comment of the reviewer. We edit the description in a simple and clear way in the revised manuscript.

  1. Please consider the correction of grammatical errors and language clarity throughout the manuscript.

Response: The manuscript is proofread by the language editing service. The present manuscript is revised.

  1. line 156, change 'companied' to 'accompanied'

Response: We apologize for the carelessness. We correct the fault in the revised manuscript.

  1. Please add some discussion about future perspectives to the findings obtained from the studies in the discussion/conclusion section

Response: We appreciate the helpful comment of the reviewer. We add some discussion about the advantage of cancer treatment through targeting FAK-induced senescence beyond many genotoxic chemotherapies-induced DNA damage and promoting senescence in non-selective cell types. It might increase the impact in the field.
